# *Morus alba* L. Cell Cultures as Sources of Antioxidant and Anti-Inflammatory Stilbenoids for Food Supplement Development

**DOI:** 10.3390/molecules30092073

**Published:** 2025-05-07

**Authors:** Vanessa Dalla Costa, Anna Piovan, Paola Brun, Raffaella Filippini

**Affiliations:** 1Department of Pharmaceutical and Pharmacological Sciences, University of Padova, Via Marzolo, 5, 35131 Padua, Italy; anna.piovan@unipd.it (A.P.); raffaella.filippini@unipd.it (R.F.); 2Department of Molecular Medicine, University of Padova, Via Gabelli 63, 35121 Padova, Italy; paola.brun.1@unipd.it

**Keywords:** *Morus alba*, calli, suspensions, stilbenoids, anti-inflammatory activity, antioxidant activity

## Abstract

*Morus alba* L. (Moraceae), white mulberry, is an ancient, well-known source of several compounds with potent biological activities and beneficial effects on human health. In this study, the juices of three stabilised undifferentiated cell lines, calli maintained in light and dark conditions, and suspensions maintained in dark condition of *M. alba* were investigated for their phytochemical content and biological activity. The results highlighted the main presence of oxyresveratrol and resveratrol-backbone glucosides, together with benzofuran derivatives. Oxyresveratrol triglucoside was found for the first time in *M*. *alba* in vitro cultures, where it represents the main compound, accounting for almost 90 µg/mL in all the juices. The total stilbenoid content resulted significantly higher in calli juices during the logarithmic phase of the growth cycle, and cell suspension juice exhibited the statistically highest total content (313.21 µg/mL of juice). Only cell suspension juice showed ROS reduction in Caco-2 cells, whereas all the juices reduced IL-1β and TNF-α levels in Caco-2 cells stimulated with LPS. These results lay the groundwork for the future exploitation of *M*. *alba* dedifferentiated cultures as sustainable resources of stilbenoid compounds to be used in the nutraceutical, cosmetic, and pharmaceutical industries.

## 1. Introduction

Plant extracts and phytochemicals have been extensively utilized in conventional medicine for the treatment of diverse diseases, making plants a valuable resource for the identification and development of novel compounds applicable in pharmaceuticals, cosmetics, and food supplements [1,2,3,4].

Medicinal plants of the Moraceae family are recognized for their diverse applications in many fields. Among over 150 species of the *Morus* genus, *Morus alba* L. (white mulberry) is the predominant one and has enormous economic value beyond sericulture [5,6].

Beyond the appreciated consumption of fruit as food, various parts of *M. alba*, including roots, stems, and leaves, together with the fruits themselves, have long been used in China, Japan, and Korea as herbal medicines due to their pharmacological effects [5,7,8,9,10]. For these reasons, the therapeutic applications of different organs or tissues of *M. alba* are reported in the Pharmacopoeia of the People’s Republic of China and also in the British Herbal Pharmacopoeia [11].

In recent years, the mulberry tree has acquired considerable importance due to the recently discovered activity of extracts derived from its various organs. This has prompted extensive phytochemical and pharmacological investigations on compounds isolated from leaves, fruits, roots, bark, root bark, and twigs [12,13]. Extracts from leaves, fruits, and seeds showed significant antioxidant activities, whereas extracts obtained especially from leaves and fruit possess anti-diabetic effects. Compounds obtained mainly from the bark, root bark and twigs are recognized for anti-inflammatory, antibacterial, antioxidant and neuroprotective properties [8,14]. Recently, researchers have identified an anti-tyrosinase activity in mulberry extracts. Indeed, the individual parts of *M. alba* have been used as a cosmetic ingredient in many Asian countries and evaluated for this activity, identifying the stilbenoid class compounds as the most important in the anti-tyrosinase activity [15,16,17]. Numerous compounds belonging to different phytochemical classes can be found in the various organs of *M. alba*, among which are alkaloids, flavonoids, stilbenoids, and, to a lesser extent, coumarins and terpenoids [9,18]. Flavonoids and stilbenoids are undoubtedly the main compounds on which attention has been focused, both classes originating from the elongation with three molecules of malonyl-CoA of the side chain of a cinnamoyl-CoA unit [13]. Among the different subclasses of flavonoids present in *M. alba* (flavones, flavonols, flavanones, dihydroflavonol, chalcones, catechins, and anthocyanidins) particular attention has been paid to the prenylated derivatives because of their great potential for biological activities [19]. In the same way as flavonoids, the stilbenoids isolated from *M. alba* stand out for their structural diversity, which differentiates them from the simple and well-known stilbenes and bibenzyls, as shown for example, by recent studies in which prenylated stilbenoids and stilbenoid Diels Alder adducts have been isolated and identified [20,21].

The large number and structural variety of these two classes of phenolic compounds present in *M. alba* will prompt efforts to obtain the phytochemicals endowed with interesting biological properties. Unfortunately, the production and accumulation of these pharmaceutically important metabolites in white mulberry trees are restricted by environmental, geographic, or seasonal factors, and the improvement of mulberry through conventional breeding is limited due to high heterozygosity and a long generation period [22]. Moreover, the extraction often causes plant death, primarily when the compounds are obtained from the root bark level.

Due to the wide range of potential health benefits, several efforts have been made to complement conventional breeding with modern biotechnological tools, mainly to increase the leaf productivity for sericulture, the compound pool production of the plant [22,23]. Most of the research has focused on protocol optimization for micropropagation and plant regeneration under in vitro conditions [24,25]. These approaches improved the secondary metabolite production, especially of rutin and stilbenoid derivatives, under different elicitation conditions [23,26,27]. However, although several published works have addressed the possible use of in vitro cell cultures of *M. alba* for the production of different metabolites, only a few papers have focused on the in vitro production of stilbenoids [28,29,30].

In this scenario, in vitro cell cultures could represent a viable and sustainable alternative to conventional methods for obtaining innovative healthy products and foodstuffs.

In a previous study [31], we described the establishment of a cell line of *M. alba* callus characterized by a sustained growth rate and a friable consistency. The preliminary chemical analyses carried out during the growth cycle (at 14, 28, and 42 days) revealed a high phenolic compound content in the callus juices, correlated to the antioxidant activity, especially in the first mid-part of the growth cycle. Moreover, seventeen compounds belonging to the stilbenoid class were identified.

In the present work, the total stilbenoid content was investigated in three *M*. *alba* undifferentiated cell lines maintained in different growing conditions: calli kept under photoperiod (M cell line), calli maintained in totally dark conditions (Md cell line), and cell suspensions maintained in the dark (Ms). The analyses were pursued on the juices extracted from the biomasses by mechanical pressing of the material to avoid the use of hazardous solvents and reduce energy and time-demanding passages. In addition, considering the demand for food products endowed with health benefits [6], the antioxidant and anti-inflammatory activities of the juices were investigated on the human intestinal Caco-2 cell line.

## 2. Results and Discussion

### 2.1. Calli and Suspensions Under Dark Conditions

Calli grown in dark conditions (Md) kept their growth rate, doubling their biomass at each growth cycle. The calli also maintained the original texture, friable and highly juicy, together with a light brown colour equal to the calli under photoperiod. Similarly, *M*. *alba* suspensions maintained in dark conditions (Ms) showed the same appearance as suspensions under photoperiod, with the same features and colours.

Growing in vitro cultures in an illuminated growth chamber requires more energy consumption than in the dark and requires technical trouble [32]. These assumptions are all the more important for the suspension cultures, which require the transition to a bioreactor system at the industrial level. Even if photoperiod is a crucial environmental factor for regulating the in vitro culture growth and development, our results highlighted the capability of *M. alba* calli and suspensions to grow in the dark, keeping the same features as the photoperiod material, without slowing biomass production.

### 2.2. Total Stilbenoid Content Expressed as Trans-Resveratrol Equivalents

Previous analyses on M juices obtained following the cell growth cycle revealed a secondary metabolite content decrease at the growth cycle end, during the late stationary phase, and, at the same time, unveiled quantitative differences in the phenolic composition between the exponential and the beginning of the stationary phase. Based on these chemical results, together with the slow growth observed in *M. alba* calli after the stationary phase, the quantitative analysis was conducted on calli grown under photoperiod (M) and under dark conditions (Md) harvested at the exponential and the beginning of stationary phases, on the 14th and 28th days, respectively. The growth cycle of in vitro cultures represents the timeframe between two subcultures; at the end, the material must be transferred into a fresh medium to replenish nutrients and eliminate potentially toxic exudates [33,34]. Concerning the suspension cultures, generally, the growth cycle is shorter because they are fast-growing cells, and, thus, the subculture intervals should be shortened. Based on the shorter growth cycle, and since either in the stationary phase or the logarithmic phase, the secondary metabolites can be mainly synthesized [33,34], juice extracted from 13-day-old cell suspension was used for the quantitative analysis during the logarithmic phase (measured by the cell volume after sedimentation method [35].

Our previous qualitative phytochemical study carried out by LC-DAD-ESI-MS on juices of *M*. *alba* callus cultures showed the predominant presence of stilbenoids and seventeen compounds were tentatively identified based on compound molecular weights, fragmentation patterns and UV spectra, further supported by the literature’s data [31]. The compounds were resveratrol derivatives (compounds **2**, **5**, **8**, **12**, **14**, **16**), oxyresveratrol derivatives (compounds **1**, **3**, **4**, **10**, **11**), and benzofuranic stilbenoid derivatives (compounds **6**, **7**, **9**, **13**, **15**, **17**) (Table 1). Then, in the present work, a quantitative analysis of the stilbenoid derivatives, expressed as trans-resveratrol equivalents, was conducted in M juices, Md juices, and Ms juice.

The absorption spectra of the stilbenoids have their maximum peak in the UV region, in the range of 300–330 nm [36], then for the analysis the chromatograms were acquired at 325, where all the identified stilbenoids are predominantly visible, and the areas under the peak were plotted against the *trans*-resveratrol standard. Figure 1A shows the chromatogram of M juice (as an example) at 325 nm, with the identified peaks numbered based on the elution time, and Figure 1B shows the relative content of each compound in the respective juice and the content (μg/mL) of the quantitatively predominant compounds is reported on the histogram. The most abundant resveratrol-backbone compounds are red-like-coloured, the most abundant oxyresveratrol-backbone compounds are blue-like-coloured, and the identified derivatives of moracin M and moracin M aglycone are green-coloured.

As appreciable from Figure 1B, oxyresveratrol glucosylated forms resulted in the most representative in all the juices; oxyresveratrol triglucoside (compound **3**) resulted the most abundant with a concentration statistically comparable (*p* > 0.05) among the calli M and Md juices, analysed on the 14th day of growth, and the suspension juice. In all these juices the concentration reached up to around 90 µg/mL. This is the first time that this compound has been found in *M*. *alba* in vitro cultures, and, as far as we know, it is absent in the plant. The concentration of oxyresveratrol glucoside (compound **10**) was approximately 86 µg/mL in Ms, representing the highest statistically significant concentration among the samples (*p* < 0.001). In Md, its concentration was approximately half that of Ms, with the concentration on day 14 being significantly higher than on day 28 (*p* < 0.05). In M, the concentration was about one-fourth that of Ms, with the day 14 concentration also significantly exceeding that of day 28 (*p* < 0.05). Additionally, small quantities of the oxyresveratrol glucoside isomer (compound **11**) and an oxyresveratrol derivative (compound **4**) were detected, along with a low amount of another unidentified oxyresveratrol derivative (compound **1**). Mulberroside A (compound **8**) was almost absent, with the statistically highest concentration (at *p <* 0.0001) in M juice on the 28th day (10 µg/mL of juice) and totally absent in Md juices. Our results contrast with some published literature in which mulberroside A was the major stilbenoid in *M*. *alba* suspension cultures. Mulberroside A, the diglucosilated form of oxyresveratrol, is considered one of the major active compounds associated with uricosoric effect, nephroprotective action, together with a strong antimelanogenic activity, present mainly at the root and root bark level [28], therefore, several attempts were done to increase its concentration in cell cultures, both at the intracellular and extracellular level. Komaikul et al. [28] stimulated *M*. *alba* cell suspension cultures with different elicitors, and they observed an increase in mulberroside A more than three-fold the control material, especially in the presence of salicylic acid, at different concentrations, and by the addition of yeast extract. The mulberroside A production was also increased by treating *M*. *alba* cell suspension with UV-C irradiation; the total mulberroside A content ranged from 16.60–21.75 mg/g dry weight in the untreated cells to 19.81–31.58 mg/g dry weight, and by adding biosynthetic precursors (L-phenylalanine and L-tyrosine), the mulberroside A content doubled the non-treated cells [37]. Even if literature data showed that the glycosylation of oxyresveratrol can enhance its activity (e.g., [38]), some authors suggested that oxyresveratrol exhibits more potent bioactivities, such as radical scavenging effects and greater anti-tyrosinase activity, thanks to its high membrane permeability, compared to its glucoside, mulberroside A, and also compared to resveratrol [39]. Moreover, in recent years, oxyresveratrol exhibited greater antioxidant activity than resveratrol, and it is considered an effective radical scavenger, beyond several other activities, including the involvement in the suppression of the inflammatory process [40,41]. In cell cultures, oxyresveratrol was found in a lower amount, about twenty times less, than mulberroside A [37].

The stilbene resveratrol represents another important structure-scaffold found in the juices. Resveratrol is a phytoalexin that possesses different physiological effects, among which are neuroprotective and cardiovascular benefits primarily due to its antioxidant effect [42]; up to now, more than 1000 stilbenoids have been isolated and identified, of which the best known are the resveratrol derivatives [43]. In our juices, we found resveratrol diglucoside (mulberroside E, compound **5**) as one of the most representative compounds, present in all the juices in concentration between 34 and 57 µg/mL, with 34 µg/mL in Md juice at 28 days (the poorest sample, at *p* < 0.005) and 57 µg/mL in M at 14 days (the richest sample, at *p* < 0.05), and also resveratrol glucoside (compound **12**), much present in Ms (32 µg/mL), followed by Md at 14th day (24 µg/mL) and only 10 µg/mL in M at 28th day (statistically different, *p* < 0.05). Compound **2**, a non-characterized resveratrol derivative, was found in all the juices. Furthermore, *trans*-resveratrol aglycone (compound **16**) was even found in low amounts in all the analyzed juices. Resveratrol was already found in *M*. *alba* cell cultures, in poorer amounts compared to the hydroxylated oxyresveratrol; its content tended to decrease after UV-C irradiation [37], whereas it increased in immobilized *M. alba* cell suspensions, the concentrations were significantly higher than those of the control group on the same day for 27- and 14-fold, surpassing the 1 mg/g of cell dry weight [29]. In all the juices, piceid (compound **14**), the resveratrol dimer, was quantified, and its concentration resulted in quite the same in all the samples.

Beyond the “classic” stilbenes, *M*. *alba* root bark, stem bark, and leaves are the main sources for arylbenzofuran derivatives, including the moracins, which sparked interest in their biological activities and their presence in natural products [44]. Moracin M (compound **17**), a naturally occurring compound in *M*. *alba* root bark, was found in all the juices, with the highest concentration (*p <* 0.0001) in Ms, 9 µg/mL of juice that is four-fold, compared to M and Md juices. Moracin M is considered one of the major compounds of *M*. *alba* twig extracts and one of the main ones responsible for twig extracts’ anti-tyrosinase activity [45]. Over the aglycone, the mono-glucosilated forms of moracin M were found as well (compounds **13** and **15**), together with the diglucosilated form mulberroside F (compound **7**). Mulberroside F, as for the other stilbenoid glucosides, showed whitening potential by inhibiting melanin synthesis, antioxidant, and anti-inflammatory effects. Mulberroside F presence was discovered in *M*. *alba* leaves, roots and in callus cultures, where it reached a concentration of 150 µg/g of dry callus extract, only slightly lower than in the root extract [30,46]. Among our juices, M ones showed the statistically highest content (*p* < 0.01) of mulberroside F, 16 µg/mL on the 14 day of the growth cycle, followed by the M juice at 28 days (12 µg/mL).

To allow a better comparison within the analysed juices, Table 2 shows the total stilbenoid contents in the juices on the 14th and 28th days of cell growth for M and Md, and on the 13th day for Ms. The total contents were derived from the sum of the single compound areas, plotted against *trans*-resveratrol standard curve (acquired at 325 nm), and expressed as *trans*-resveratrol equivalents (µg/mL of juice).

Regarding the juices of the calli M and Md, the data indicate that the total content of stilbenoids is significantly higher during the logarithmic phase of the growth cycle, and no significant differences were highlighted between calli maintained in light and calli maintained in the dark. The data clearly indicate that the juice of Ms obtained from cell suspensions kept in the dark has the highest stilbenoid content, statistically superior to the juices M and Md obtained from calli (*p* < 0.001). It should be emphasized that suspensions, in general, are the in vitro material necessary for achieving scalability at the industrial level; the results obtained, which demonstrated the superiority of suspensions over calli regarding biosynthetic capabilities, are particularly interesting also because the analyzed suspensions are cultivated in total darkness, and this is also a significant parameter since cultivation in a bioreactor rarely allows exposure of the cells to light [47].

The results highlighted the preferential biosynthetic pathway of *M*. *alba* in vitro cultures in synthesizing stilbenoid compounds compared to the qualitative composition of other *M*. *alba* plant parts. Stilbenoids, majorly found in plant roots, root bark, and twigs, are scarcely present in the leaves, in which only oxyresveratrol was found as stilbene, in addition to several moracins [48]. Kim et al. [49] reported a total stilbenoid content of about 189 mg/100 g DW (dry weight) in leaves and 609 mg/100 g DW in fruits. In any case, in the studies above, the stilbenoids represent only a part of the secondary metabolite composition of the plant extract, which is composed of several other molecules like acid phenols, flavonoids, anthocyanins, and coumarins [48,49]. *M. alba* in vitro cultures obtained in this work underlined the possibility of using this material as a preferential source of stilbenoids to be used in nutraceuticals, cosmetics, and pharmaceutical sectors.

### 2.3. Identification of Non-Toxic Concentrations of Juices

Thinking about the dietary consumption as food supplements of juices obtained from *M. alba*, in this study, we considered the human intestinal epithelial Caco-2 cell line as an in vitro model for studying the biological activity of the juices. At first, the non-toxic concentrations of juices were assessed by MTT assay in Caco-2 cells incubated for 16 h with juices at concentrations ranging from 0 to 50% *v*/*v*, 1:2 serial dilutions. We reported toxic effects in cells at the lower concentration of 0.7812% *v*/*v* for M and Md (Figure 2). Our data are in line with previous reports indicating no reduction in cells viability in a human liver cell line up to 0.5 mg/mL fruit extract [50]. With the aim to compare the biological effects of M, Md, and Ms, we decided to perform subsequent experiments at 0.39% *v*/*v*, as mean the higher non-toxic concentration of Md and Ms. As reported above, *M. alba* callus juices are a mixture of several stilbenoids together with other different molecules not identified in this work, therefore, the concentrations of juices were reported as % *v*/*v*. However, keeping in mind the data reported in Table 2, it is possible to infer that 0.39% *v*/*v* corresponds to 80 ng stilbenoid in M, 88 ng in Md, and 129 ng in Ms.

### 2.4. Juices from Suspension Cultures Slightly Reduced Intracellular ROS Generation

Physiological levels of oxygen free radicals are involved in cell signalling and defence mechanisms. However, increased levels of reactive oxygen species (ROS) determined, for example, by bacterial infections, result in cell damage and exhaustion of the cellular antioxidant capabilities. Indeed, oxidative stress is common in many chronic diseases [51]. In this study, we tested the ability of juices to quench the ROS generation induced in Caco-2 cells. Because of the polyphenols’ relatively short half-life [52], we pre-treated Caco-2 cells with extracts twice.

As reported in Figure 3, in cells challenged with H_2_O_2,_ intracellular ROS levels increased by 5.8-fold compared to non-treated cells (*p* < 0.0001). Even if the fruit from different *Morus* species extracts showed ROS scavenging activity mainly attributed to the high phenolic compositions [53,54], in our experiments, only suspension juice (Ms) showed significant effects (*p* < 0.0001) when compared to H_2_O_2_ treated cells. Indeed, the M and Md lines did not reduce ROS production in H_2_O_2_-stimulated cells, whereas Ms reduced by 1.84-fold intracellular ROS levels. The antioxidant activities of *M. alba* extracts have already been reported. As an example, Jiang DQ and colleagues reported that products purified from mulberry fruits reduce oxidative stress in mice experimentally subjected to physical exercise [55].

Interestingly, while all extracts should possess antioxidant potential due to their phenolic content, only the suspension juice (Ms) demonstrated a significant reduction in H_2_O_2_-induced intracellular ROS levels, suggesting that specific compositional, synergistic effects among the components or bioavailability differences may influence the effectiveness of juices obtained from undifferentiated cell cultures in counteracting oxidative stress in cells.

### 2.5. Effects on Lipopolysaccharide-Induced Inflammation

Intestinal epithelial cells are frequently exposed to bacterial products such as lipopolysaccharide (LPS). LPS from pathogens triggers an inflammatory cascade leading to inflammation, dysbiosis, and intestinal dysmotility. In this study, we first ruled out the possibility that juices induced a pro-inflammatory effect on Caco-2 cells. As reported in Figure 4, panels A and B, Caco-2 cells incubated for 24 h with extracts did not report an increase in IL-1β and TNF-α production compared to non-treated cells. Upon stimulation with LPS, Caco-2 cells significantly increased IL-1β and TNF-α production compared to non-treated cells (*p* < 0.0001). M, Md, and Ms significantly diminished the LPS-induced pro-inflammatory cytokine generation (Figure 4, panels C and D); moreover, we did not detect significant differences among the cell line juices, meaning that all the biomasses preserved the anti-inflammatory capability. At the same, Li and colleagues reported that ethanolic extracts with different chemical moieties obtained from the branch bark of *M. alba* reported comparable anti-inflammatory activities on RAW cells [56]. Moreover, ethyl acetate and acetone extracts from root bark of *M. alba* reported significant reduction in nitric oxide production (a pro-inflammatory mediator) with IC_50_ values in the order of 10 µg/mL [16].

Altogether, our data suggest that juices obtained from *M. alba* dedifferentiated cultures possess anti-inflammatory properties capable of attenuating LPS-induced cytokine production in intestinal epithelial cells, highlighting their potential as food additional products useful in modulating gut inflammation without eliciting a basal pro-inflammatory response.

## 3. Materials and Methods

### 3.1. In Vitro Callus and Suspension Cultures

Calli of *M*. *alba* (M) were already obtained from aseptically germinated seedlings (Saflax, Münster, Germany), cultured on Murashige and Skoog’s basal medium [57] containing 1 mg/L dichlorophenoxyacetic acid, supplemented with 30 g/L sucrose, solidified with agar (8 g/L), and the pH was adjusted at 5.7 before autoclaving; the material was maintained at 25 °C ± 1 in a 16/8 h photoperiod chamber. After seventeen subcultures, part of the stabilized calli was moved to dark conditions (Md), in the same media used for parental calli and at 25 °C ± 1. Suspensions were established by transferring a suitable quantity of M calli into Erlenmeyer flasks containing the same liquid medium as the solid material. Suspension flasks were kept at 100 rpm and 25 °C under 16/8 h photoperiod and subcultured every 14–32 days. After five cultures, the stabilized suspensions were transferred into dark conditions (Ms) at 100 rpm and 25 °C ± 1 (Figure 5).

### 3.2. Juice Preparation

Based on the preliminary phytochemical analyses and the biomass growth, juices were obtained from calli maintained both under photoperiod and in dark conditions harvested on the 14th and 28th days of the growth cycle, and from cell suspensions grown in dark conditions harvested on the 13th day.

Calli were taken from Petri dishes and maintained at −18 °C. The cells from suspension cultures were filtered and washed with distilled water; then, the suspension cells were left to dry until they lost the surrounding water and were maintained at −18 °C. The biomasses were squeezed after defrosting, adding quartz powder. The ground samples were sonicated for 40 min, and after centrifugation at 13,200 rpm, the supernatant of each sample (juice) was taken and analyzed. The juices were named as the material they derived from: M and Md for the juices extracted from *M*. *alba* calli grown under photoperiod and dark conditions, respectively, and Ms for the juice from *M*. *alba* cell suspensions maintained under photoperiod.

### 3.3. Chemicals

HPLC-grade methanol and acetonitrile, and analytical-grade ethanol and acetic acid were purchased from VWR-BDH Chemicals (Milan-Italy). Ultrapure water was obtained using a Sartorius Arium system (Sartorius Italy, Varedo, Italy). *Trans*-resveratrol reference standard was purchased from Merck (Milan, Italy). Basal medium components, vitamins, hormones, and agar for the preparation of *M. alba* cell culture media were purchased from Duchefa (Micropoli, Milan, Italy).

### 3.4. Quantitative Analysis of Stilbenoids Expressed as Trans-Resveratrol Equivalents

The quantitative analysis was performed by HPLC-DAD using an Agilent 1100 HPLC Series System (Agilent, Santa Clara, CA, USA) equipped with a degasser, a quaternary gradient pump, a column thermostat, and a diode array detector. A C6-Phenyl Gemini column (5 µm, 250 × 4.6 mm) from Phenomenex (Torrance, CA, USA) was employed, and the column temperature was maintained at 40 °C throughout the analysis. The mobile phase consisted of 0.15% acetic acid in water (A) and acetonitrile (B), with the following gradient elution program: 97% A at 0–6 min, 75% A at 15 min, 75% A at 20 min, 20% A at 30 min, and 97% A at 40 min. The flow rate was 1 mL/min, with an injection volume of 10 µL; chromatograms were acquired at 325 nm [58].

The content of stilbenoids was expressed as *trans*-resveratrol equivalent using an authentic commercial standard. Resveratrol standard solution (1 mg/mL) was prepared in ethanol, and the calibration curve was obtained in a concentration range of 1.5–125 µg/mL, with six concentration levels. Peak areas were plotted against corresponding concentrations (y = 80.268x − 21.156, R^2^ = 0.999). The analysis was performed in duplicate, and the results were expressed as mean ± standard deviation (SD).

### 3.5. Cell Culture and Cell Viability

In this study, the human epithelial cell line Caco-2 (ATCC^®^ HTB-37™, LGC Standards; Milan, Italy) was used to study the biological effects of juices obtained from *M. alba*. Indeed, the Caco-2 cell line is a well-characterized, reproducible and easy-to-culture in vitro model for studying the anti-inflammatory effects of compounds thought to be introduced with the diet or as supplements [59]. Caco-2 cell line was maintained in Eagle’s Minimum Essential Medium supplemented with 20% heat-inactivated fetal calf serum and 100 U/mL penicillin (all provided by Thermo Fisher Scientific; Milan, Italy). Cells were seeded at 4 × 10^4^ cells/mL in cell culture plates (Corning^®^ Costar^®^ provided by Merck, Milan, Italy) and incubated at 37 °C, 5% CO_2_. Cultured media were renewed 24 h later. At 95% confluency, cells were incubated with juices prepared as described in 4.2.

For cell viability assay, Caco-2 cells were seeded in 96 well culture plates (4 × 10^4^ cells/mL, 100 µL). The growth media were renewed every 24 h and 48 h later (cell confluency at 95%), cells were incubated with juices for 16 h. Cells were then added of 5 mg/mL MTT (3-(4,5-dimethylthiazol-2-yl)-2,5-diphenyltetrazolium bromide, Merck; Milan, Italy) solution. Cultures were incubated for additional 4 h at 37 °C. The resulting formazan crystals were then dissolved in 10% *w*/*v* sodium dodecyl sulfate 10% containing 0.01 N HCl. The absorbance was measured 16 h later at 590 nm using a microplate reader (MultiPlateReader VictorX2, Perkin Elmer; Milan, Italy).

### 3.6. Intracellular Reactive Oxygen Species Measurement

Caco-2 cells were seeded in 24 well culture plates (4 × 10^4^ cells/mL, 500 µL). The growth media were renewed every 24 h and 48 h later (cell confluency at 95%), cells were incubated with juices for 16 h. At the 15th hour of incubation, stimuli were renewed. Thirty minutes later, cells were loaded in warm PBS with 50 μM 2′,7′-dichlorodihydrofluorescein diacetate (H_2_DCFDA) probe (Molecular Probes, Invitrogen; Milan, Italy) and incubated in the dark for 30 min at 37 °C. At the cytoplasmic compartment, H_2_DCFDA is converted by intracellular esterases and oxidized to fluorescent 2′,7′-dichlorofluorescein (DCF). The generation of DCF is directly proportional to cytoplasmic ROS levels [60]. Cells were challenged with 25 μM H_2_O_2_ (Merck). Ten minutes later, cells were washed, harvested using Trypsin-EDTA, and analyzed using BD FACSCantoTM flow cytometry system, collecting ten thousand events. Results were analyzed using the Floreada.io software (https://floreada.io/analysis; accessed on 17 December 2024).

### 3.7. Enzyme-Linked Immunosorbent Assay

Caco-2 cells were seeded in 96 well culture plates (4 × 10^4^ cells/mL, 100 µL). The growth media were renewed every 24 h, and 48 h later (cell confluency at 95%) cells were incubated for 16 h with extracts in the presence or not of 100 ng/mL lipopolysaccharide (LPS) from *Salmonella enterica* serotype Typhimurium (Merck). At the end of incubation, conditioned media were collected. Levels of interleukin (IL)-1β and tumour necrosis factor (TNF)-α were measured in the conditioned media using commercially available enzyme-linked immunosorbent assay kits (ELISA, Thermo Fisher Sci), reporting analytical sensitivity of 2 pg/mL and 0.13 pg/mL, respectively. Assays were developed using 3,3,5,5-tetramethylbenzidine (TMB). Data were recorded at 450 nm using a microplate reader (MultiPlateReader VictorX2, Perkin Elmer).

### 3.8. Statistical Analysis

Chemical results were reported as mean ± standard deviation (SD) of two independent experiments, each performed in duplicate, and biological results as mean ± standard error of the mean (SEM) of three independent experiments, each performed in triplicate. Statistical analysis was performed using one-way analysis of variance (ANOVA) using GraphPad Prism v 7.05 (San Diego, CA, USA); *p* values < 0.05 were considered statistically significant. In the chemical investigations, different Latin letters (which denote significant differences at *p* < 0.05) are attributed in alphabetic order from the richest samples to the poorest samples.

## 4. Conclusions

This study evaluated juices obtained from three cell lines of *M. alba* in vitro cultures. The results indicate that juices, which represent a product obtainable without solvents and therefore with low environmental impact, have potential in terms of chemical composition and biological activity. Although cell lines were cultured without elicitation, they demonstrated a good biosynthetic potential in the production of stilbenoids and promising antioxidant and anti-inflammatory activity. To obtain material that can be used in both pharmaceutical and food applications, future developments will mainly focus on identifying low-cost and low-impact precursors and elicitors that are compatible with their use in human nutrition. Moreover, our study paves the way for the scaling-up of stilbenoids production using cell lines. Indeed, cell culture systems provide a controlled and reproducible environment, enabling the consistent production of specific molecules independent of seasonal and geographical variables. The transition from laboratory to industrial-scale production poses significant challenges. Technical limitations such as shear sensitivity of cells, oxygenation of the cultures and nutrient transfer inefficiencies in large bioreactors can hinder productivity. Additionally, the cost of establishing and maintaining sterile culture conditions and media, as well as regulatory considerations related to food and pharmaceutical applications may represent limitations. Despite these limitations, the standardisation of the obtained products and the economic advantages of the cell line production highlight the potential for sustainable and targeted botanical production.

## Figures and Tables

**Figure 1 molecules-30-02073-f001:**
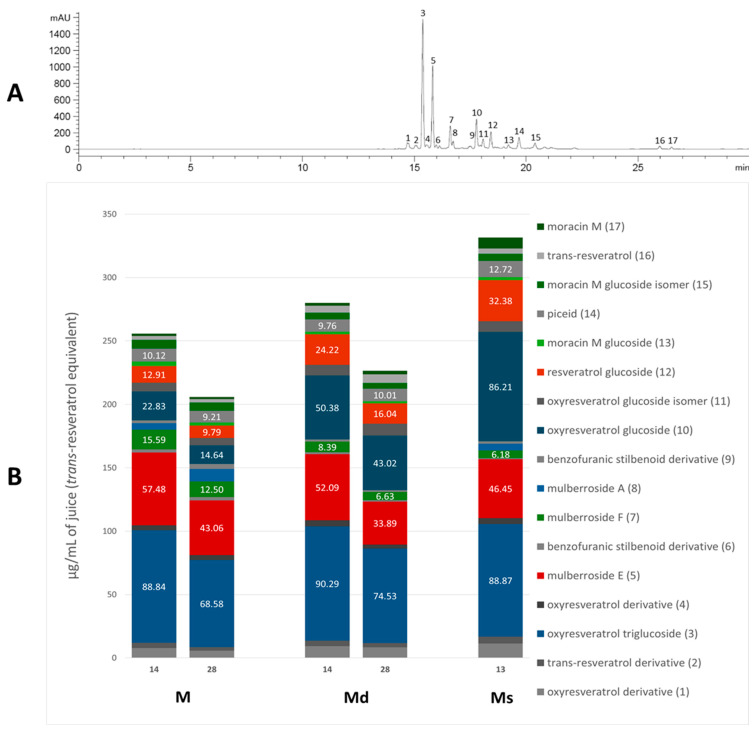
Chromatogram of M juice, acquired at 325 nm, with the identified peaks numbered (panel (**A**)); single stilbenoid (numbered as in the chromatogram) amount in M and Md on the 14th and 28th days of the growth cycle, and Ms juice on the 13th day of growth cycle (panel (**B**)).

**Figure 2 molecules-30-02073-f002:**
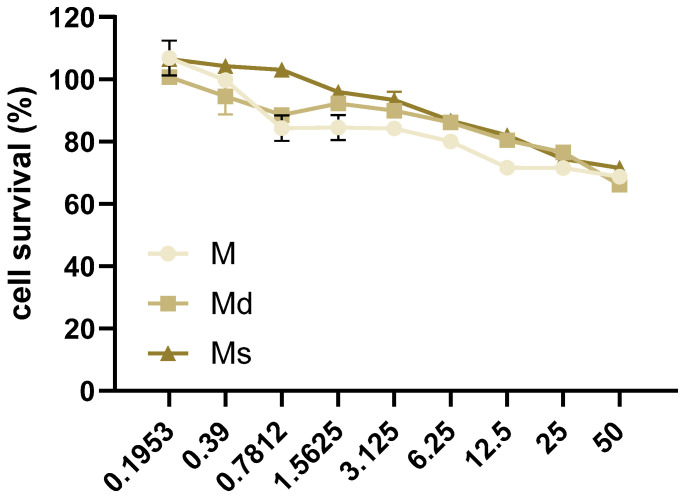
Caco-2 cells were incubated with juices at 50–0% *v*/*v* with 1:2 dilutions for 16 h. Cell survival was determined by MTT assay and reported as the percentage (%) of living cells calculated over cells incubated with juices 0% *v*/*v*. Data are reported as mean ± SEM of three experiments.

**Figure 3 molecules-30-02073-f003:**
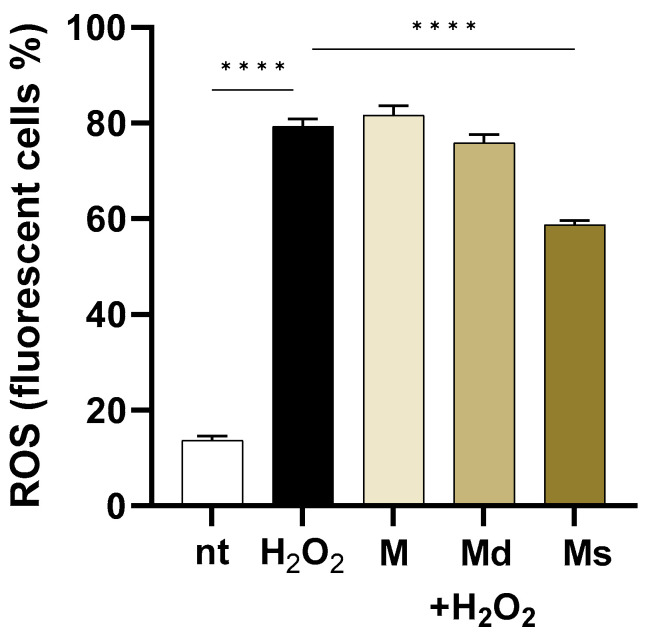
Caco-2 cells were incubated with juices at 0.39% *v*/*v* and then loaded with H_2_DCFDA. Intracellular ROS were detected by flow cytometry in 10^4^ cells. Data are reported as the percentage of fluorescent positive cells expressed as mean ± SEM of three experiments. **** indicates *p* < 0.0001. nt: non-treated cells.

**Figure 4 molecules-30-02073-f004:**
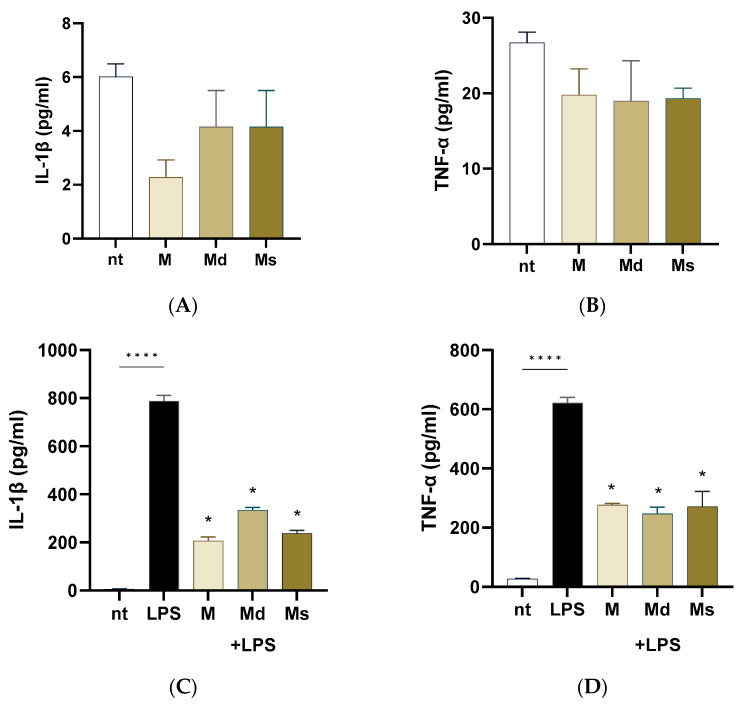
Caco-2 cells were incubated with LPS 100 ng/mL with or without juices for 16 h. Cytokine levels were assessed on the conditioned media using ELISA kits. (**A**) IL-1b measured in the conditioned media of cells not treated with LPS. (**B**) TNF-alpha levels measured in the conditioned media of cells not treated with LPS. (**C**) IL-1b measured in the conditioned media of cells treated with LPS. (**D**) TNF-alpha levels measured in the conditioned media of cells treated with LPS. Data are reported as mean ± SEM of three experiments. **** indicates *p* < 0.0001; * indicates *p* < 0.0001 vs. LPS-treated cells. nt: non-treated cells.

**Figure 5 molecules-30-02073-f005:**
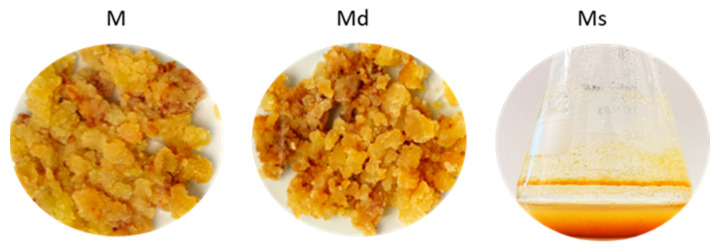
Callus grown in photoperiod (M), in dark conditions (Md), and suspension culture grown in dark conditions (Ms).

**Table 1 molecules-30-02073-t001:** Backbone structures of the isolated stilbenoid compounds.

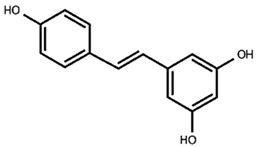 resveratrol backbone
**2**—*trans*-resveratrol derivative
**5**—mulberroside E
**12**—resveratrol glucoside
**14**—piceid
**16**—*trans*-resveratrol
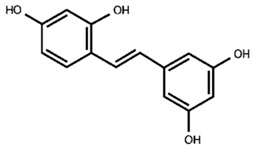 oxyresveratrol backbone
**1**—oxyresveratrol derivative
**3**—oxyresveratrol triglucoside
**4**—oxyresveratrol derivative
**8**—mulberroside A
**10**—oxyresveratrol glucoside
**11**—oxyresveratrol glucoside isomer
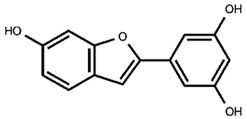 benzofuran stilbenoid backbone
**6**—benzofuranic stilbenoid derivative
**7**—mulberroside F
**9**—benzofuranic stilbenoid derivative
**13**—moracin M glucoside
**15**—moracin M glucoside isomer
**17**—moracin M

**Table 2 molecules-30-02073-t002:** Total stilbenoid content (*trans*-resveratrol equivalents) of M, Md and Ms juices. Results are expressed as mean ± SD, n = 2. The significant differences at *p* < 0.05 are denoted by different Latin letters.

Cell Line	Growth Cycle (Days)	Total Stilbenoid Content (µg/mL of Juice)
M	14	255.71 ± 2.22 ^b^
	28	205.92 ± 3.26 ^c^
Md	14	279.94 ± 7.47 ^b^
	28	226.56 ± 2.19 ^c^
Ms	13	331.55 ± 5.40 ^a^

## Data Availability

The presented data are available on request from the authors.

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
