# Peer review of "Morus alba L. Cell Cultures as Sources of Antioxidant and Anti-Inflammatory Stilbenoids for Food Supplement Development"

_molecules, 2025, doi:10.3390/molecules30092073_

Round 1

Reviewer 1 Report

Comments and Suggestions for Authors

Introduction: 

L 35-43: It seems to me that such a detailed description of the plant itself is not necessary in this case. The article is intended for scientists, who certainly have knowledge about the appearance and general use of white mulberry.

L 78: this part should be removed: (Lee et al., 2011; Inyai et al., 2021; Sabater-Jara et al., 2023). 

L 78-80: I would not agree with this opinion completely. There are many papers on the possibility of using in vitro M. alba cell cultures for the production of metabolites, but indeed few specifically concern stilbenoids. In my opinion, it should be changed to make it clear that it concerns stilbenoids.

L 95: this should be removed (Jan et al., 2021),

Results and Discussion: 

L 99: 2.1. Calli and suspensions under dark conditions. 

Are there any pictures available that would allow a schematic overview of the experiment? 

L 113-115: Cell growth can be defined as size, biomass, cell number or metabolite content, and during this period, called growth cycle, cells face four phases: lag, exponential (or logaritmic), stationary and death phases [30].

I think that there is no need to describe the cell growth stages in this article at this level.

L 126-127: As for suspension cultures, the growth cycle is usually shorter because they are rapidly growing cells and therefore the intervals between subcultures [31].....

Please finish or change this expression. 

L 176: Approximately the same concentration of oxyresveratrol glucoside, compound 10 (86 175 µg/mL of juice), was found in Ms, whereas its concentration was pretty half in Md and one-fourth in M. 

Pretty should be removed. 

Materials and Methods: 

L 361-362: Could the authors add 1-2 sentences about why the juices were obtained on days 14 and 28 and from cell suspensions harvested on the 13th day? 

Juices were obtained from calli harvested on the 14th and 28th day of the growth cycle  and from cell suspensions harvested on the 13th day. 

References: 

Ozturk, M.; Azra, N.; Kamili, A.N.; Altay, V.; Rohela, G.K.; Morus alba In: Mulberry - From Botany to Phytochemistry; Springer 474 
Nature Switzerland AG, 2024; pp. 7-9. https://doi.org/10.1007/978-3-031-49117-7 

Mohamad Puad, N.I., Abdullah, T.A. (2018). Monitoring the Growth of Plant Cells in Suspension Culture. In: Multifaceted Pro-532 
tocol in Biotechnology; Amid, A.; Sulaiman, S.; Jimat, D.; Azmin, N. Eds. Springer, Singapore, 2018; pp. 203-214. 533 
https://doi.org/10.1007/978-981-13-2257-0_17 

Kumar, P.P.; Loh, C.S. Plant tissue culture for biotechnology. In: Plant Biotechnology and Agriculture Prospects for the 21st Century; 535 
Altman, A.; Hasegawa, P.M.P. Eds. Academic Press, 2012; pp. 131-138. https://doi.org/10.1016/B978-0-12-381466-1.00009-2 536 

In my opinion, these citations should be removed or replaced with the newer one:

Bourgaud, F.; Gravot, A.; Milesi, S.; Gontier, E. Production of plant secondary metabolites: A historical perspective. Plant Sci. 2001, 537 

Chattopadhyay, S.; Farkya, S.; Srivastava, A.K.; Bisaria, V.S. Bioprocess Considerations for Production of Secondary Metabolites 539 
by Plant Cell Suspension Cultures. Biotechnol. Bioprocess Eng. 2002, 7, 138–149. https://doi.org/10.1007/BF02932911 
161, 839–85. https://doi.org/10.1016/S0168-9452(01)00490-3 

Comments on the Quality of English Language

The quality of the English language should be checked by a native speaker or a qualified person.

Author Response

Dear Editor,

We would like to thank You and the Reviewers for time and effort to review the manuscript.

We revised the text according to the suggestions.

Reviewer 1

Introduction

All changes have been made as indicated.

Results and discussion:

2.1. Are any pictures available that would allow a schematic view of the experiment?

We added the pictures in the Material and Methods section for a better visualisation of the biomasses.

All the other points: the Reviewer is right and we followed all the suggestions.

References

Done

Reviewer 2 Report

Comments and Suggestions for Authors

The article "White mulberry cell cultures as sources of stilbenoids for food supplement development" presents an interesting approach to obtaining bioactive phytocompounds from in vitro plant cell cultures, preserving the plant itself and overcoming issues related to fruit seasonality.

Comments:

Introduction: Flavonoids and stilbenoids are mentioned, but no chemical background is provided. It would be beneficial to expand the introduction by discussing the chemical classification of these molecules.

Materials and Methods: The section lacks a description of the chemicals used.

Specific comments on methodology:

Line 383: The reference for the supplier of the standard is missing.

Line 387: If possible, the equation of the calibration curve should be included. Moreover, the quantitative analysis is performed using only one standard representing the entire molecular class, making it a relative quantitative analysis. The title of section 2.2 should be adjusted accordingly.

Line 390: Caco-2 cells are epithelial cells derived from human colorectal adenocarcinoma. The manuscript should correctly contextualize these cells in the Materials and Methods section and throughout the manuscript.

Line 393: Specify where the cells were seeded (Petri dishes, plates?).

Line 394: The term monolayer refers to a differentiated state of Caco-2 cells, which occurs after 21 days of culture when they transition from colonocytes to intestinal epithelium. A more appropriate term should be used to describe cell confluency, along with clearer details on the confluency state at the time of treatment.

Cell culture experiments (sections 3.5 and 3.6): These sections are unclear and lack details on cell seeding, growth conditions, and confluency status before treatment. Additionally, supplier information (e.g., Merck) is missing.

HPLC-DAD analysis:

Line 132: A table should be included listing the molecules used as a database for the quantitative HPLC-DAD analysis.

Line 138: The 325 nm wavelength corresponds to the maximum absorption (λmax) of the standard used in the calibration curve. The explanation in lines 132-144 is unclear and should be rewritten for better clarity.

Figure 1: How were the different stilbenes identified in the chromatogram if no standards were used and the only detector was DAD, which does not provide specific information for metabolite identification?

Lines 142-144: Including an image illustrating the structural classification of stilbenes would improve clarity.

Line 173: How was the identity of the detected metabolite confirmed? The presence of a previously unreported metabolite should be confirmed through isolation, purification, and structural analysis (e.g., NMR) or by comparison with a commercial standard. If this has not been done, one of these approaches should be implemented before claiming its identification.

Cell viability and ROS assays:

Section 2.3: The cell line description should be corrected, and concentrations should be expressed consistently with the Materials and Methods section (instead of using dilution percentages). Additionally, the study does not include a large amount of data, so adding MTT assay results would enhance the robustness of the findings. MTT experimental data should be presented.

Lines 297-300: The conclusion that the biological activity of the juices is due to a significant polyphenol content is not supported by experimental data. This is based only on previous literature, without confirming the presence and amount of polyphenols in the tested juices. A quantitative assessment of total polyphenols should be performed using HPLC-DAD or the Folin-Ciocalteu method. Several protocols available in the literature can be easily implemented to complete this analysis.

Line 315: If intracellular ROS levels were measured using flow cytometry, the methodology should be described in the Materials and Methods section.

Conclusions and industrial feasibility:

Lines 440-443: The study aims to demonstrate that in vitro cultures can provide a sustainable source of phytochemicals for food, nutraceutical, and pharmaceutical applications. However, there is no discussion of potential scale-up and industrialization of the process. A dedicated paragraph should be added, or this topic should be discussed in the conclusions, addressing the feasibility, advantages, and/or limitations of scaling up the proposed approach into an industrial facility. This would strengthen the conclusions, which currently appear to me too concise.

Comments on the Quality of English Language

The manuscript is not written in fluent English, and numerous grammatical and lexical errors suggest that it has not been proofread by a native or proficient speaker. A deep language revision is strongly recommended (examples: lines 32, 54, 176, 178, 376-377, 396-401).

Author Response

Dear Editor,

We would like to thank You and the Reviewers for time and effort to review the manuscript.

We revised the text according to the suggestions.

Reviewer 2  

Introduction:

Flavonoids and stilbenoids are mentioned, but no chemical background is provided. It would be beneficial to expand the introduction by discussing the chemical classification of these molecules.

You are right, thank You. We improved the chemical background of the mentioned classes in the Introduction.

Materials and Methods:

The section lacks a description of the chemicals used.

We added the suppliers and the equation of the calibration curve, and modified the title of section 2.2.

Line 390, 393, 394, sections 3.5 and 3.6.

As You suggested, these parts have been revised and some paragraphs have been better described.

HPLC-DAD analysis:

Line 132: A table should be included listing the molecules used as a database for the quantitative HPLC-DAD analysis.

Figure 1: How were the different stilbenes identified in the chromatogram if no standards were used and the only detector was DAD, which does not provide specific information for metabolite identification?

Line 173: How was the identity of the detected metabolite confirmed? The presence of a previously unreported metabolite should be confirmed through isolation, purification, and structural analysis (e.g., NMR) or by comparison with a commercial standard. If this has not been done, one of these approaches should be implemented before claiming its identification.

As stated in the text, the identification has been reported in “Dalla Costa, V.; Piovan, A.; Brun, P.;

Filippini, R. Morus alba calli: a sustainable source of phytochemicals and nutritive supplements. In

vitro Cell. Dev. Biol. Plant 2025, submitted” Ref. 31.

The identification was carried out through LC-MS/MS and HPLC/DAD analyses based on compound molecular weights, fragmentation patterns, and UV spectra, further supported by literature data. In the work, all the data are reported and well discussed, and we deemed it appropriate not to report them again.

Line 138: The 325 nm wavelength corresponds to the maximum absorption (λmax) of the standard used in the calibration curve. The explanation in lines 132-144 is unclear and should be rewritten for better clarity.

Done

Lines 142-144: Including an image illustrating the structural classification of stilbenes would improve clarity.

Done.

Cell viability and ROS assays.

Done

Conclusions and industrial feasibility

The Reviewer’s suggestion is well-taken. The section “Conclusions” has been revised.

Reviewer 3 Report

Comments and Suggestions for Authors

The significance and scientific foundation of the study are solid, and the topic is interesting, particularly because it concerns a species that is unfortunately becoming increasingly rare worldwide, both due to climate change and human factors. I would also like to praise the study design. However, corrections and additions need to be made as I have outlined in detail:

Title

  • I suggest adding the Latin name of the plant species to the title.
  •  I also suggest changing the entire title, as it does not fully reflect the scope of the research. In addition to the phytochemical analysis, the study includes an investigation of antioxidant and anti-inflammatory activity.

Introduction

  • Although the focus of the study is on the chemical composition, the Introduction section contains very little information on this aspect, particularly regarding stilbenoids. It is necessary to provide a more detailed discussion on this topic.
  • Additionally, there are multiple repetitions concerning the biological activity and medicinal benefits. These sections should be revised and merged to improve clarity and conciseness.

Results and Discussion

  • Why are HPLC chromatograms not presented for all three samples in the Results section? It would be beneficial to include them for completeness and better data interpretation.
  •  I suggest presenting Figure 1B as a table for better clarity. In its current form, it is quite confusing to identify the specific quantitative content being reported.
  • Considering that Molecules is a chemistry-focused journal, you should include the structures of the main identified metabolites. These structures should be drawn using ChemDraw or another relevant program for chemical structure representation.
  • It is necessary to explicitly state in the text whether the key comparative results showed statistically significant differences or not.

  • In subsection Quantitative Analysis (line 125), it is necessary to number the reference properly and adjust the numbering of all subsequent references accordingly.

  • In the subsection Identification of Non-Toxic Concentrations of Juices and the results, there is no comparison with the findings of other authors, nor is there a discussion of these results.
  • The subsection Effects on Lipopolysaccharide-Induced Inflammation needs to include a comparison with previous studies. You should discuss whether your findings confirm or contradict the results and claims of other authors.

Materials and Methods

  • You have not consistently provided manufacturers for standards, reagents, solvents, and instruments (including model details).
  • Please ensure that all materials and equipment used in the experimental section are properly referenced with their respective manufacturers and model specifications.
  • Please include the calibration curve equation used for resveratrol quantification along with its correlation coefficient to ensure transparency and reproducibility of the results.

Conclusion

  • I recommend expanding the conclusion to include a summary and conclusions for each segment of the experiment conducted. This will provide a clearer overview of the key findings and their implications.

Author Response

Dear Editor,

We would like to thank You and the Reviewers for time and effort to review the manuscript.

We revised the text according to the suggestions.

Reviewer 3

Title

The title has been modified according to Your suggestion

Introduction

The Reviewer is correct. As suggested, we improved the part regarding stilbenoids and revised the part on biological activity and medicinal benefits.

Results and Discussion

Why are HPLC chromatograms not presented for all three samples in the Results section? It would be beneficial to include them for completeness and better data interpretation.

We reported only one chromatogram because the profile of the different chromatograms is very similar, with only a few differences in the area of the peaks, so we thought their presentation was redundant.

I suggest presenting Figure 1B as a table for better clarity. In its current form, it is quite confusing to identify the specific quantitative content being reported.

It is necessary to explicitly state in the text whether the key comparative results showed statistically significant differences or not.

You are right, the table would be preferable in terms of completeness of data, however we believe that the visual impact of histograms is more immediate. We have decided to plot the contents of the most significant quantitative compounds on histograms. We also reported the statistical significance data in the text.

Considering that Molecules is a chemistry-focused journal, you should include the structures of the main identified metabolites. These structures should be drawn using ChemDraw or another relevant program for chemical structure representation.

We completely agree with your suggestion. We inserted a Table with the main three backbones and the related compounds.

In subsection Quantitative Analysis (line 125), it is necessary to number the reference properly and adjust the numbering of all subsequent references accordingly.

Done

In the subsection Identification of Non-Toxic Concentrations of Juices and the results, there is no comparison with the findings of other authors, nor is there a discussion of these results.

We discussed the results and compared the data with published findings.

The subsection Effects on Lipopolysaccharide-Induced Inflammation needs to include a comparison with previous studies. You should discuss whether your findings confirm or contradict the results and claims of other authors.

Done

Materials and Methods

You have not consistently provided manufacturers for standards, reagents, solvents, and instruments (including model details).

Done

Please ensure that all materials and equipment used in the experimental section are properly referenced with their respective manufacturers and model specifications.

Done

Please include the calibration curve equation used for resveratrol quantification along with its correlation coefficient to ensure transparency and reproducibility of the results.

Done

Conclusions

I recommend expanding the conclusion to include a summary and conclusions for each segment of the experiment conducted. This will provide a clearer overview of the key findings and their implications.

Done

The main changes are highlighted in red in the text.

Round 2

Reviewer 2 Report

Comments and Suggestions for Authors

The overall quality of the manuscript has significantly improved, hence I have no further comments. 

Reviewer 3 Report

Comments and Suggestions for Authors

I would like to express my appreciation to the authors for carefully considering and incorporating my suggestions and comments to revise and enhance the manuscript.